# Beyond Reconstruction: Self-Supervised Representations for Brain Tumor Molecular Subtyping

**Max Van Puyvelde**[*1,2]                                                    MAXVPUYV@STANFORD.EDU
**Ibrahim Gulluk**[*3]                                                            GULLUK@STANFORD.EDU
**Wim Van Criekinge**[2,†]                                            WIM.VANCRIEKINGE@UGENT.BE
**Olivier Gevaert**[1,†]                                                      OGEVAERT@STANFORD.EDU

[1] *Department of Biomedical Data Science, Stanford University School of Medicine*

[2] *Department of Mathematical Modelling, Statistics and Bioinformatics, Ghent University*

[3] *Department of Electrical Engineering, Stanford University*

[†] *Co-senior authors*

## Abstract

Both masked autoencoders (MAE) and convolutional autoencoders such as AutoencoderKL (AKL) learn to reconstruct voxels, but they differ in architecture and training strategy: MAE uses global self-attention on heavily masked input, while AKL uses local convolutions on complete input. Does this difference affect whether the resulting representations capture clinically relevant structure? We compare frozen ViT-Large MAE features to AKL on two glioma molecular subtyping tasks (IDH1 mutation, WHO grade) across 1,110 multi-site glioma MRI volumes, sweeping the per-token projection width from 32 to 1,152 (38K–1.4M total latent dimensions). We find that MAE features outperform AKL by up to 8.2 AUC points for IDH1 at every compression level, including at matched total dimensionality. Already at $16\times$ compression ($d'=32$, 38K total dims), MAE surpasses AKL at its full capacity (614K dims), suggesting that masking combined with global self-attention preserves clinical structure under compression far more efficiently than local convolutional features.

**Keywords:** Self-supervised learning, Masked autoencoder, Representation learning, Brain MRI, Vision Transformer, Glioma

## 1. Introduction

Convolutional autoencoders such as AutoencoderKL (AKL) (Rombach et al., 2022) are the dominant learned tokenizer for 3D medical image generation (Pinaya et al., 2022). Both AKL and masked autoencoders (MAE) (He et al., 2022) are ultimately trained to reconstruct voxels, but they differ in two key respects: (1) the MAE must reconstruct from heavily masked input, forcing it to encode global context, while AKL sees complete volumes; (2) the MAE encoder relies on global self-attention while AKL uses local convolutions. This distinction matters for diffuse markers like IDH1 mutation, a molecularly defined hallmark of diffuse glioma (Cancer Genome Atlas Research Network, 2015; Louis et al., 2021) whose subtle imaging signature has motivated dedicated MRI-based prediction methods (Choi et al., 2021) and may require whole-volume context to detect. We study whether these architectural and training differences translate into better clinical representations by freezing a pretrained MAE ViT-Large encoder and projecting its features to varying dimensionalities $d' \in \{32, \ldots, 1152\}$ (Figure 1).

---

[*] Equal contribution

Table 1: IDH1 mutation and WHO grade prediction (AUC, 5-fold CV). LR: logistic regression; RF: random forest. MAE outperforms AKL across all compression levels. Best in **bold**. ‡Matched total dimensionality (614K).

| Encoder | $d'$ | Dims | IDH1 ($n$=1,012) | | Grade ($n$=495) | |
|---|---|---|---|---|---|---|
| | | | LR | RF | LR | RF |
| AKL | 8 | 614K | $.801_{\pm.032}$ | $.746_{\pm.053}$ | $.793_{\pm.083}$ | $.653_{\pm.065}$ |
| MAE | 32 | 38K | $.861_{\pm.017}$ | $.810_{\pm.029}$ | $.791_{\pm.035}$ | $.755_{\pm.058}$ |
| | 128 | 154K | $.877_{\pm.026}$ | $.846_{\pm.019}$ | $.827_{\pm.015}$ | $.781_{\pm.039}$ |
| | 256 | 307K | $.876_{\pm.031}$ | $.856_{\pm.012}$ | $\mathbf{.851}_{\pm.037}$ | $.809_{\pm.060}$ |
| | 512‡ | 614K | $.865_{\pm.023}$ | $.859_{\pm.022}$ | $.839_{\pm.030}$ | $.817_{\pm.060}$ |
| | 1,152 | 1.4M | $\mathbf{.883}_{\pm.020}$ | $\mathbf{.868}_{\pm.022}$ | $.846_{\pm.030}$ | $\mathbf{.825}_{\pm.070}$ |

## 2. Method

**Data.** T1c brain MRI from UPenn-GBM (Bakas et al., 2022) (615 patients) and UCSF-PDGM (Calabrese et al., 2022) (495 patients), totaling 1,110 volumes. All skull-stripped, registered to SRI atlas, resampled to 160×192×160, and z-score normalized.

**Self-supervised pretraining.** A ViT-Large (Dosovitskiy et al., 2021) (12 layers, $d$=1152, 16 heads, ∼304M params) is pretrained with 70% masked image modeling for ∼400 epochs, following self-supervised 3D medical pretraining practice (Tang et al., 2022). Input is patchified ($16^3$) into $N$=1,200 tokens. After pretraining, the encoder is frozen.

**Projection sweep.** The frozen encoder maps $\mathbf{x} \mapsto \mathbf{Z} \in \mathbb{R}^{N \times 1152}$. Per-token linear projections $f_\theta : \mathbb{R}^{1152} \to \mathbb{R}^{d'}$ for $d' \in \{32, 128, 256, 512, 1152\}$ are each paired with a CNN decoder (residual blocks, self-attention, 4× upsampling; MSE loss). Only the projection and decoder are learned.

**Baseline.** AKL (Rombach et al., 2022): channels $[64, 128, 256]$, two stride-2 downsamplings (4× spatial), 8 latent channels, 8×40×48×40 = 614,400 total dims. Trained from scratch on the same data with L1 loss.

**Evaluation.** For each encoder and $d'$, we mean-pool features across tokens and evaluate via 5-fold patient-stratified cross-validated linear probing (logistic regression, $C$=1.0; random forest, 500 estimators). Tasks: **IDH1 mutation** ($n$=1,012; both cohorts, patients with available IDH1 status) and **WHO grade** 4 vs. 2/3 ($n$=495, PDGM only).

## 3. Results

Table 1 presents the central finding. For IDH1 mutation, MAE outperforms AKL at every compression level, with the gap widening from +6.0 at $d'$=32 to +8.2 AUC at full dimensionality (LR). Even at 16× compression ($d'$=32, 38K total dims), MAE already surpasses AKL at its full capacity (614K dims). At matched dimensionality ($d'$=512, both 614K), MAE achieves 0.865 vs. 0.801, confirming the advantage stems from representation quality rather than capacity. For WHO grade ($n$=495), MAE surpasses AKL from $d'$=128 onward (LR), with continued gains up to $d'$=256 (0.851 vs. 0.793).

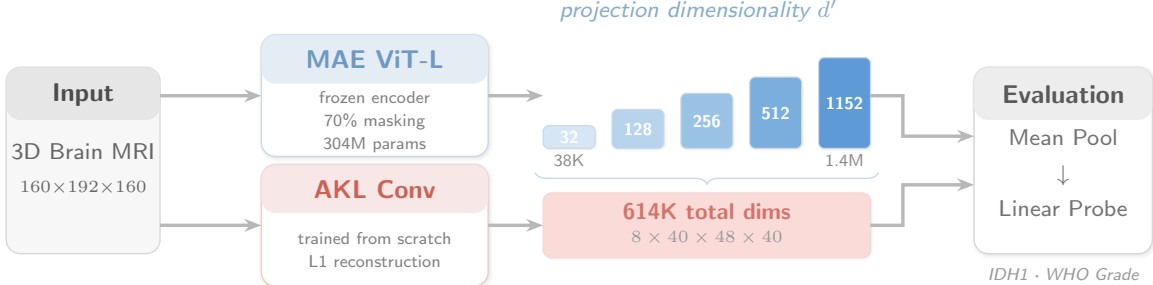

Figure 1: Experimental setup. Both encoders map the same volumes to latent representations that are mean-pooled and linearly probed. The MAE path sweeps projection dimensionality $d'$ to trace how clinical signal distributes; the AKL baseline operates at a fixed 614K total dimensions.

## 4. Discussion

Two findings emerge. First, at matched total dimensionality (614K), MAE still outperforms AKL by 6.4 AUC points for IDH1, confirming the advantage is not simply one of capacity. Second, MAE features remain competitive with AKL even under extreme compression: at $d'{=}32$ (38K dims, $16\times$ fewer than AKL), the MAE already exceeds AKL's full-capacity IDH1 performance. We attribute this to the combination of masking and self-attention: reconstructing heavily masked volumes via global attention forces the encoder to learn brain-wide patterns relevant to diffuse molecular markers, whereas convolutional encoders trained on complete input can rely on local statistics.

**Limitations.** Both encoders are trained on 1,110 volumes from two glioma cohorts (T1c only), ensuring a fair comparison of training paradigms rather than data access. The AKL baseline uses the widely adopted configuration from Rombach et al. (channels $[64, 128, 256]$, 8 latent channels); sweeping AKL hyperparameters was not feasible due to computational constraints.

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
