# OpenReview forum: "Beyond Reconstruction: Self-Supervised Representations for Brain Tumor Molecular Subtyping"
_MIDL.io/2026/Short_Papers — MIDL 2026 - Short Papers Poster_

### Official Review · Reviewer_R6Fd · 2026-04-23
**interesting comparison of distinct autoencoding architectures**

**Rating:** 4
**Confidence:** 4

**Review:**

The authors investigate the question of how the type of architecture of autoencoder influences their capability of extracting information about IDH mutation and for the WHO classification of brain tumors. There are two fundamentally different architectures that are both popular and widely used. The paper confirms an intuition of MAEs handling tasks for which global context is important better.

**Summary:**

Autoencoders are universal tools for feature extraction. In this paper two completely different types of autoencoder architectures are compared in the task of extracting clinically relevant features in the tasks of brain tumor molecular subtyping. On the one hand the Masked Autoencoder (a transformer based architecture) as well as an AutoencoderKL (a convolutional architecture), both widely used models.

**Strengths:**

The paper is clearly written and easy to understand. It investigates a fundamental question for this type of tasks, and the experiments are well thought out. The authors do address and discuss certain limitations.

**Weaknesses:**

The whole study uses mainly default configurations for the two models which is on the one hand a good baseline, on the other hand it makes the claim abut the model type weaker, as the lower performing AKL could be a result of the suboptimal choice of certain hyperparameters. However it is a difficult to make a “fair” comparison in that regard. In this light the choice of the widely used default parameters is an acceptable limitation.

For the actuall AUC in table 1 I would have expected a monotonic relationship between dimensionality and AUC. This is however not the case for all the experiments, which could indicate a greater variability, or the need for more data. However there is a visible trend.

**Justification Of Rating:**

The paper proposes an interesting comparison of two high performing but completely different autoencoder encoder architectures, and provides some evidence that certain properties of MAEs are favourable for the task at hand.

---

### Decision · Program_Chairs · 2026-05-08

Accept (Poster)